# Polarization and Moral Threat: Insights from Systemist Analysis

Ilona Goldner * and Pazit Ben-Nun Bloom

Department of Political Science, The Hebrew University of Jerusalem, Mount Scopus, Jerusalem 9190401, Israel; pazit.bennun@mail.huji.ac.il
* Correspondence: ilona.gregor@mail.huji.ac.il

**Abstract:** This article presents a theoretical framework for understanding the social consequences of polarization-driven behaviors by conceptualizing them as a moral threat to the self. Our argument employs systemist graphics, illustrating key connections and patterns from two distinct scientific works. First, an analysis of polarization-driven behavior, which reveals Americans' willingness to trade democratic values for partisan goals. Second, research on moral disengagement strategies, revealing the role of resentment as a coping mechanism in armed conflicts. We offer a synthesis analysis between these two studies and uncover a twofold role of morality in polarization: as a factor in forming partisan animosity and a catalyst in its perpetuation and intensification. We further highlight the role of outgroup hate, rather than ingroup love, in driving negative actions resulting from polarization, and the challenge of reconciling morally-driven conflicts. Our framework sheds new light on the complex interplay between morality and conflicts, with implications for social cohesion, erosion of moral values, and democratic backsliding.

**Keywords:** affective polarization; conflict resolution; democracy; moral disengagement; morality

## 1. Introduction

Affective polarization—a term in political psychology referring to the tendency to perceive opposing partisans negatively and co-partisans positively—has risen in the United States in recent decades (Iyengar et al. 2019). The literature on affective polarization consistently shows its negative social consequences, such as day-to-day social distancing and threatening political behavior (Iyengar et al. 2019). However, the literature has often overlooked the moral psychology mechanisms that underlie social conflicts. In this article, we present a theoretical framework that argues that morality plays a two-fold role in affective polarization, both as establishing partisan animosity and catalyzing and reinforcing it, thereby generating a self-reinforcing cycle of hostility. Our argument draws from two distinct interdisciplinary fields. Firstly, we draw from affective polarization studies which emphasize the fundamental consequences of polarization for liberal democracy and the risk to democratic norms (Kingzette et al. 2021). Secondly, we draw from moral psychology literature describing moral disengagement strategies and the function of resentment as a coping strategy that reduces moral threat and self-condemnation (Bandura et al. 1996).

We use systemist graphics to combine these different kinds of literature. By simplifying complex articles' arguments within each study, the systemist graphic method allows for uncovering the logical inferences and connections between the two studies with straightforward graphic notation. This highlights the need to consider moral psychology mechanisms when studying affective polarization and its consequences. The text in each systemist figure is typed in UPPER- or lower-case characters. UPPER case characters are used for MACRO variables, while lower case characters are used for micro-level variables. Each figure also comes in double frames—the outer refers to the environment, and the inner refers to the system. In both studies, an academic discipline (Political Science or International Relations)

is the system, and the World Beyond is its environment. The macro level of Political Science (and International Relations) is the discipline as a whole, and the micro level is individual scholars within it (for an extensive explanation of the method and its notation, see Gansen and James (2023)).

In the current article, we use two functions of systemist graphics. First, we create a systemist graphic for two articles to clarify their arguments and the literature they lay upon. Graham and Svolik's (2020) work illustrates how polarization can drive political behavior, showing that Americans are willing to trade off their democratic values for partisan goals by voting for politicians who have violated democratic principles. While this highlights the potential danger of democracy backsliding as a result of polarization, their work does not address the psychological implications of such a trade-off between personal values. Ben-Nun Bloom et al.'s (2020) work is then used to explain moral disengagement strategies, particularly the role of resentment and trivialization as coping strategies after morally threatening decision-making in armed conflicts. Secondly, using systematic synthesis, we combine the findings of the two articles and suggest that resentment towards the outgroup can intensify as a coping mechanism for moral threat. Finally, we highlight two critical implications of associating moral disengagement with harsh acts encouraged by affective polarization, stressing the difficulty of reconciling conflicts driven by moralization. This analysis exposes the intricate ways in which moral psychology mechanisms can amplify social animosity in diverse contexts, including in both domestic and international spheres.

## 2. Polarization, Moralization, and Support for Liberal Democracy

From Schumpeter's "minimalist" conception of procedural democracy to the more substantive definitions that encompass essential freedoms to ensure meaningful elections, one constant characteristic of democracy is the competition between political groups (Collier and Levitsky 1997). Accordingly, ideological political polarization, which reflects the degree to which a society diverges along ideological lines, is, to some extent, inherent in the process of democratic competition and fundamental to the essence of democracy itself. However, in recent years, scholars have shifted the focus from ideological polarization to its typically subsequent, non-ideological, affective polarization. This new type of polarization uncovers the growing emotional divide between individuals with different political views, and a heightened level of mistrust or hatred towards people on the opposite side of the political spectrum (Iyengar et al. 2012). Affective polarization is considered to be rooted in partisanship, as a strong social identity, leading to political "us versus them" social dynamics. This classic ingroup–outgroup distinction is heightened by changes in the political and social environment (e.g., political identity alignment and the rise of partisan media), leading to today's often extreme levels of partisan animosity (for a review of the origins of affective polarization, see Iyengar et al. 2019).

While affective polarization is a worldwide phenomenon, it is often considered to be stronger today in the United States than in any other country (Boxell et al. 2022). Thus, most of the literature about polarization is based on American politics, including the extensive studies documenting the frequently non-political consequences of increased affective polarization. These consequences can include personal preferences, such as avoiding romantic relationships with out-partisans (Huber and Malhotra 2017). They can also take the form of discriminatory economic behaviors, such as paying those from a different political party less money for the same work (McConnell et al. 2018) or exhibiting bias during the hiring process by favoring political ingroup members (Gift and Gift 2015).

In the case of increased out-party animosity, Finkel et al. (2020) have identified three core elements of affective polarization: othering, aversion, and moralization. They theorize that even when none of these elements individually poses a significant threat to democratic societies, the combination of the three can have dire consequences for liberal democracy, especially with the wholesale moralization of political parties. In this matter, adopting a moralized identification with one's own political group and against opposing partisans can create a sense of perceived existential threat, in which the thought of political loss becomes

unacceptable, leading to an enduring need to prevent it at any cost. This understanding of the role of moralization in enhancing animosity follows the conclusions of other morality studies, which show that when individuals hold a firm moral conviction on a political issue, they are more likely to make harsher judgments regarding that issue (Ben-Nun Bloom 2014). With the relatively recent tendency in the United States for conflict between political parties to appear as an existential threat, the direct political consequences of affective polarization have only recently begun to unfold (Iyengar et al. 2019).

The most troubling political consequence of affective polarization is the undermining of democratic norms and electoral accountability (Kingzette et al. 2021). This effect manifests through several concerning outcomes, including the spread of political fake news (Osmundsen et al. 2021), induced tolerance to corrupted political practices (Anduiza et al. 2013), and the active support for political candidates who flout democratic norms (Graham and Svolik 2020). While scholars debate whether affective polarization truly leads to decreased support for democratic norms (Broockman et al. 2022), the examples provided demonstrate that even when public support for these norms is high, affective polarization can undermine the motivation to uphold them and hold politicians accountable, especially when partisan power is at stake. Referring to Schmitter and Karl's (1991) definition of democracy, which emphasizes it as a system of governance where citizens hold rulers accountable through competition, affective polarization poses a direct threat to true electoral competition. In essence, even if the public cherishes democratic values, polarization can heighten the risk of democracy's backsliding by challenging its core characteristic—competitive governance that is accountable to the people. Graham and Svolik's (2020) work presents a notable example of such reasoning and behavior, an argument we will now lay using systemist visual representation.

Figure 1 displays Graham and Svolik's (2020) article in a systemist graphic form. As the article addresses the political implications of affective polarization, the system of the diagram is "Political Science", and the environment is "World Beyond", presenting the variables and implications outside the discipline. There are 14 "variables" covering both the macro level of Political Science as a discipline and the micro level of individual scholars' work within it.

The diagram starts with the initial variable "Through democratic elections, the public serves as a democratic check", which presents Graham and Svolik's overall observations about competitive democratic elections, mainly the role of such events as the basis of democratic accountability. To this, they add the complexity behind voter behavior, indicating how "electoral competition often confronts voters with a choice between potentially conflicting considerations". The possibilities involved validate their argument that "polarization might affect the price of prioritizing democratic principles over partisan interests". Due to the existential threat affective polarization generates, the perceived stakes of winning elections are increased. Next, the authors, as we put it, "test the willingness to trade off democratic principles for partisan interests", using two studies, a "candidate-choice experiment" and one natural experiment that occurred during Montana's 2017 special election for the U.S. House of Representatives. In the candidate-choice experiment, "candidates' attributes were manipulated (including partisan affiliation and policy platforms), while some politicians took positions that violate key democratic principles"; this tested voters' willingness to trade off democratic principles for partisan interests, and to distinguish between considerations of preferred policy versus partisan affiliation. These logically possible results are then represented separately as the willingness to trade off democratic principles for either partisan loyalty or preferred policies, a distinction related to the difference between ideological and affective polarization.

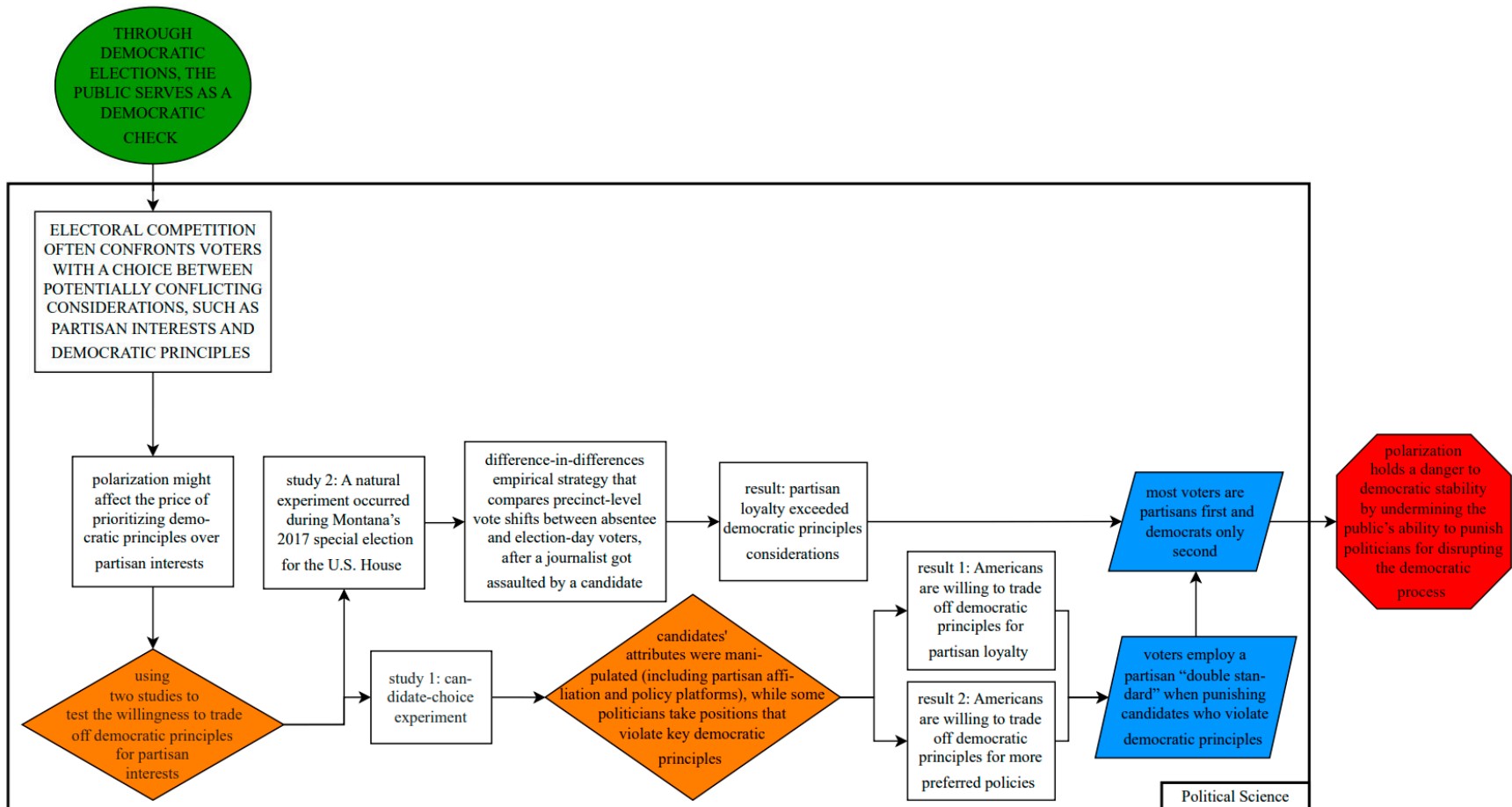

**Figure 1.** Diagram of the article "Democracy in America? Partisanship, Polarization, and the Robustness of Support for Democracy in the United States" (Graham and Svolik 2020). The diagram provides a comprehensive overview of the article's main argument. It starts with the theoretical foundation of democratic elections and their role in ensuring democratic accountability. Next, it highlights the methods used to support the hypothesis that polarization hampers voters' ability to prioritize democratic values over partisan interests. The red octagon serves as a visual representation of the article's central finding, indicating that polarization may increase the risk of democratic backsliding. Diagrammed by: Pazit Ben-Nun Bloom, Ilona Goldner, Sarah Gansen and Patrick James.

Following on, the results of Graham and Svolik show that Americans are willing to trade off democratic principles for both considerations. These possibilities converge on the conclusion that "voters employ a partisan 'double standard'" for "candidates who violate democratic principles". The respondents are not indifferent to democratic norms; they possess them, but they are willing to relinquish them for partisan goals, a point that will become critical later when we synthesize the two schematically presented arguments. We follow Graham and Svolik's general term "Americans" without distinguishing the judgment of Democrats from Republicans, as the authors did not identify any partisan asymmetry in the willingness to compromise democratic norms for partisan ends.

Study 2 is then used as a robustness check for these findings. In the Montana natural experiment study, the authors applied a difference-in-differences empirical strategy to compare precinct-level vote shifts between absentee and election-day voters, after a candidate assaulted a journalist. The researchers inferred that "partisan loyalty exceeded democratic principles" since only moderate precincts voted across party lines, and thus prioritized partisan ends over democratic norms. The final argument concludes that "polarization holds a danger to democratic stability by undermining the public's ability to punish politicians for disrupting the democratic process", meaning that polarization weakens and threatens commitment to democratic electoral procedures and norms.

While Graham and Svolik do not treat variations or different levels of affective polarization, they do highlight the tendency toward prioritizing the goals of the ingroup, such as winning an election, over other significant values, such as democratic norms. This suggests the ability of affective and ideological polarization to undermine a democratic political system, at least if they reach a sufficient level, while spiraling with growing animosity and conflict fueled by moralization. However, this does not mean that the choice of some voters to downgrade democratic principles is easy, as these may be values they still hold along with those that eclipse them in importance. In different words, we suggest that moral threat might arise from the collision between two values: the security of their political ingroup versus the value of fairness in liberal democracy. This raises the question of the long-term social effects of harsh acts performed in the name of partisan conflict. To tie in the discussion to moral threat and seek an answer to this question, we now turn to the literature on moral values and voters' engagement or disengagement with them.

## 3. Moral Disengagement in Conflicts

The importance of morality to self-identity is a major finding in the literature on morality in psychology (Aquino and Americus 2002); it is generally regarded as the most valuable attribute of one's identity, and its loss is the most devastating (Strohminger and Nichols 2014). Morality also plays a crucial role in shaping social interactions. This is demonstrated by the importance of ingroup morality over competence and sociability as a source of pride and identification (Leach et al. 2007), and in social judgments when forming impressions about others (Brambilla et al. 2011). Thus, morality is vital for maintaining a positive self-identity and ingroup image. Consequently, immoral behavior may threaten one's moral identity and undermine the sense of self-worth, generating feelings of stress and self-condemnation. This judgment of behavior as immoral is not grounded in an objective truth regarding the fundamental nature of morality, as debated in philosophy on the concept of the "absolute good". Instead, it is subjective to each individual, influenced by their internal moral standards and values. In this sense, morals are often perceived as inherently guiding actions and even serving as motivations in their own right (Ben-Nun Bloom 2013). Thus, negative self-judgment resulting from a threat to moral identity highly motivates individuals to repair their moral reputation or employ coping mechanisms (Pagliaro et al. 2016). Such coping mechanisms may include heightened moral behavior for compensatory reasons (Jordan et al. 2011) and physical cleansing, which synthesizes moral purity and body purity (Zhong and Liljenquist 2006). By engaging in these behaviors, individuals may seek to counteract the negative impact of a moral threat to their identity.



Another common strategy for facing moral threat is moral disengagement, which Bandura et al. (1996) define as a set of cognitive tactics allowing people to bypass self-condemnation by disengaging their morally questionable acts from their internal standards and moral self-views. Originally, Bandura's moral disengagement theory was thought to explain moral judgment prior to immoral actions, showing how individuals diminish the moral self-regulation mechanisms that would ordinarily prevent them from engaging in wrongful behavior. However, developments in this theory suggest that moral disengagement should be conceptualized as a multistage process (Tillman et al. 2018), serving as a coping mechanism that reduces self-condemnation after an immoral act has occurred. This complements Bandura's original identification of four kinds of disengagement strategies based on the aspect of a person's moral "self-regulatory system" that they undermine: those related to the morality of the conduct, personal responsibility reduction, misrepresenting detrimental consequences, or victim-orientated strategies.

Two additional points should be noted while recognizing the importance of moral disengagement as a coping mechanism. First, individuals are influenced by the actions of their ingroup. Therefore, an immoral ingroup member's actions may lead to a moral threat due to a symbolic threat to the group's image (Brambilla et al. 2013). To protect their moral group identity (and, by extension, self-identity), people may employ similar coping mechanisms to those used in response to personal moral threats. Thus, for example, Täuber and Zomeren (2013) found that moral threats to the national ingroup identity elicited more outrage toward outgroup nations.

Secondly, a moral threat does not solely result from direct immoral actions but can also arise from negative self-judgment based on moral considerations. This includes situations where individuals fail to meet moral standards, such as dilemmas with conflicting moral principles. Ben-Nun Bloom et al. (2020) developed this argument by applying Monin's (2007) social comparison theory, which suggests that moral threat can result from social comparison when we observe others acting more morally than ourselves. Following this, Monin suggested three strategies people may use to defuse this moral threat: suspicion, trivialization, and resentment. These resemble Bandura et al.'s (1996) four groups of disengagement strategies, as suspicion downplays the moral implications of the good deed, trivialization the consequences of it, and resentment focuses on the hostility toward the target. Central to both theories is how negative feelings toward others help reduce the moral threat to the self.

Ben-Nun Bloom et al. (2020) use these implications to explain the social outcomes of moral disengagement in an ongoing armed conflict, where moral threats arise from repeated moral dilemmas of the ingroup. The article shows how trivialization and resentment form a spiral of moral disengagement that gradually motivates harsher moral decisions, leading to a lessening moral backbone for the society in conflict. To explain this argument most clearly, we provide a systemist visual representation of the article in Figure 2.

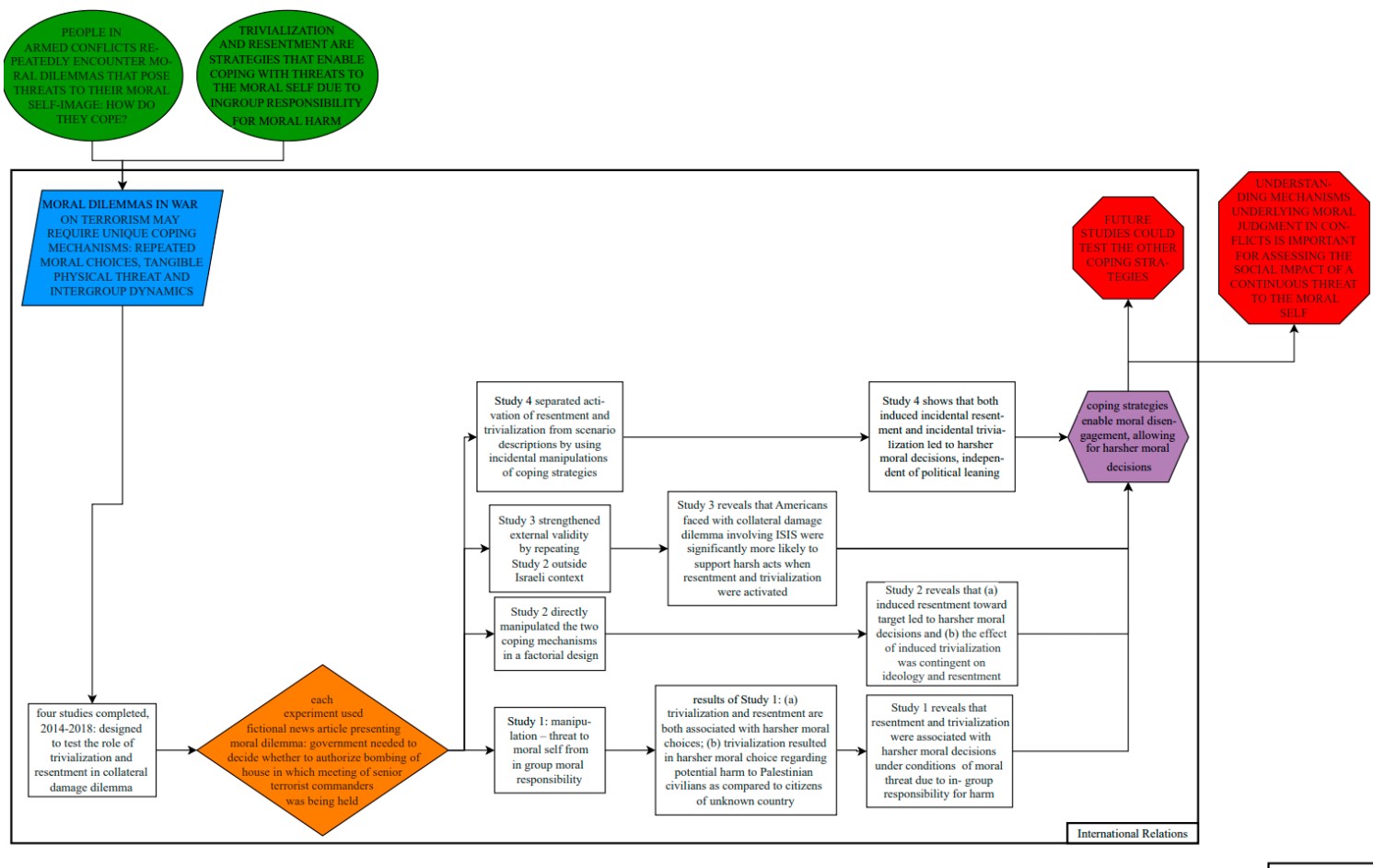

**Figure 2.** Diagram of the article "Coping with Moral Threat: Moral Judgment Amid War on Terror" (Ben-Nun Bloom et al. 2020). The diagram offers a comprehensive overview of the main argument in the article. It starts with the theoretical conceptualizations of armed conflicts dilemmas as moral threats to the self, and presents trivialization and resentment as potential coping mechanisms. Next, the diagram outlines the four experiments presented in the article, showing how these coping mechanisms with moral threats enable moral disengagement. This ultimately leads to the red octagon, which visually represents the article's central conclusion: continuous moral threat may weaken the social moral backbone. Diagrammed by: Pazit Ben-Nun Bloom, Ilona Goldner, Sarah Gansen and Patrick James.

As the article addresses coping mechanisms for a moral threat that arises from asymmetric war moral dilemmas, the chart concerns the "system" of "international relations" and its environment in the "world beyond". Seventeen variables cover both the macro level of International Relations as a discipline, and the micro level of individual scholar's work within it. The initial question of how people in armed conflicts cope with dilemmas that threaten their moral self-image leads, following literature on moral disengagement, to identifying trivialization and resentment as possible coping strategies for threats to moral self-image in the instance of ingroup responsibility for moral harm. In the context of an asymmetric war on terror, which is overwhelmingly stressful and threatening to the moral self, unique coping mechanisms seem to be required.

The researchers conducted four experiments "designed to test the role of trivialization and resentment in collateral damage dilemmas". As indicated, "each experiment used a fictional news article presenting a moral dilemma", involving a case in which "the government needed to decide whether to authorize the bombing of a house in which a meeting of senior terrorist commanders was being held". The first and third studies deal with the Israeli–Palestinian and US–ISIS conflicts, respectively, while the second manipulates the two coping mechanisms involved in the first study, trivialization and resentment, to determine their role as causes, and the fourth looks separately at the two strategies with the aid of manipulations using incidental primes.

In the diagram, each study description is followed by its result. Study 1 suggests that "trivialization and resentment result in harsher moral choices", particularly "under conditions of moral threat due to ingroup responsibility for harm". The results of Study 2 confirm this and add that "the effect of induced trivialization was contingent on ideology and resentment". Study 3 found that "Americans faced with a collateral damage dilemma involving ISIS were significantly more likely to support harsh acts when resentment and trivialization were activated". Study 4 found that induced incidental resentment and trivialization both led to harsher moral decisions, independent of political leanings. It was then hypothesized that trivialization and resentment facilitate moral disengagement by reducing self-condemnation in retrospective thinking, allowing for harsher moral decisions in future dilemmas. It was concluded, generally, that "understanding mechanisms underlying moral judgment in conflicts is important for assessing the social impact of a continuous threat to the moral self".

With this conclusion from the moral psychology literature, we can now return to our initial question about the long-term social effects of harsh moral acts performed in the name of partisan conflict. This can be conducted using what, in systemist graphics, is called "systematic synthesis", a technique explained further in the introductory article for this special issue (Gansen and James 2023).

## 4. Systematic Synthesis

At first glance, the articles of Graham and Svolik (2020) and Ben-Nun Bloom et al. (2020), may seem profoundly dissimilar, as they belong to different academic disciplines and areas of focus. However, the use of systemist graphics to illustrate their arguments highlights that they address a shared underlying social phenomenon, with the main difference lying in the contexts of the conflicts they examine.

While this is not explicitly stated, Graham and Svolik's work identifies a moral dilemma arising from the trade-off between the goals of the partisan ingroup, such as winning an election, and other values that individuals with it likely hold, such as democratic norms. Amid the increased affective polarization and animosity between political parties, such a conflict creates tension between the two principles that can be difficult to reconcile, and, thus, may create a moral threat to the self. The diagram in Figure 3a incorporates this inferred connection into the original systemist diagram of the article. The dotted line, which symbolizes a linkage inferred by us and not explicitly made by the original authors, leads to a new terminal variable: "the trade-off between partisan goals and democratic norms is a moral dilemma that may create a moral threat to the self".

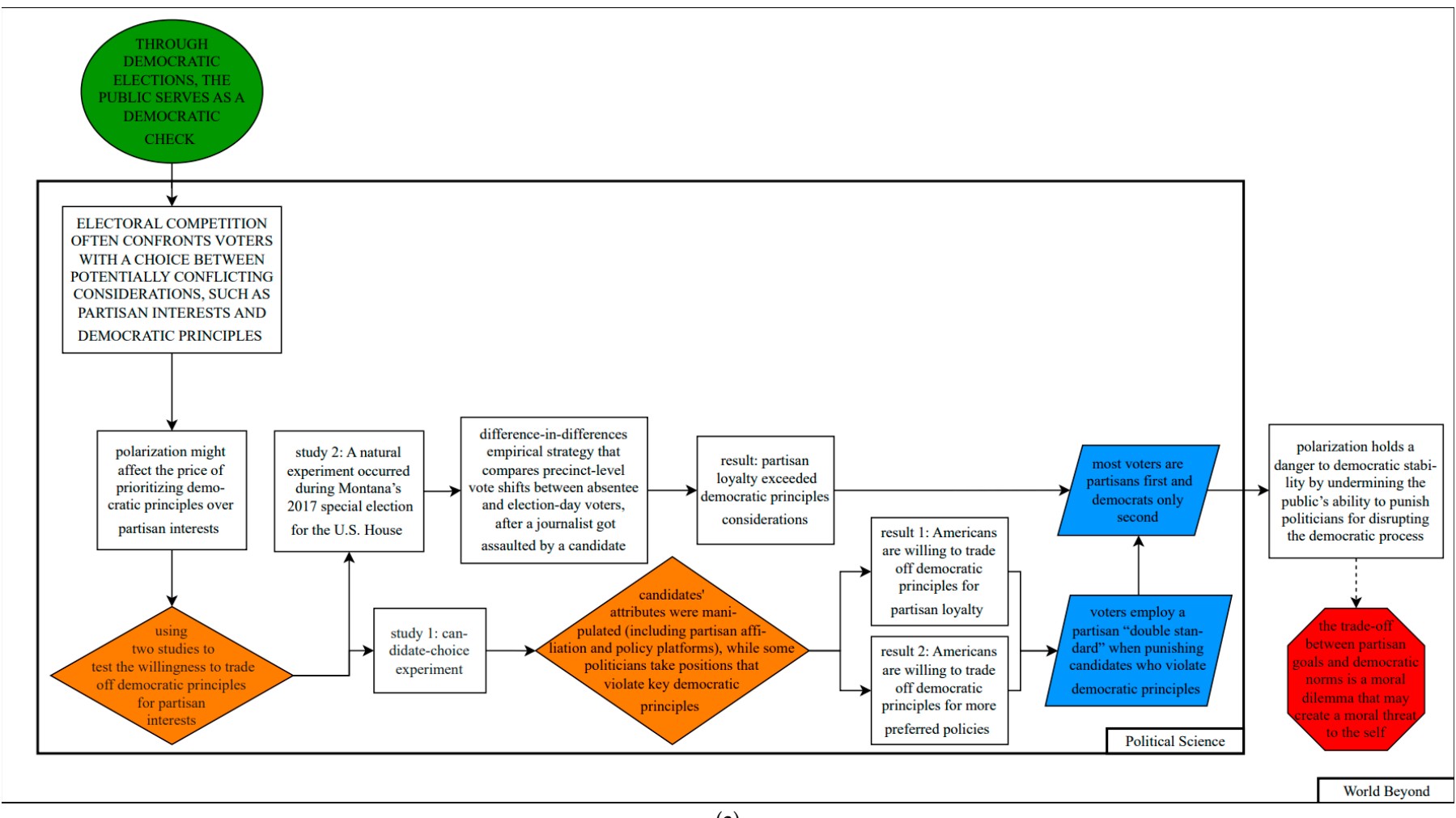

(**a**)

**Figure 3.** *Cont.*

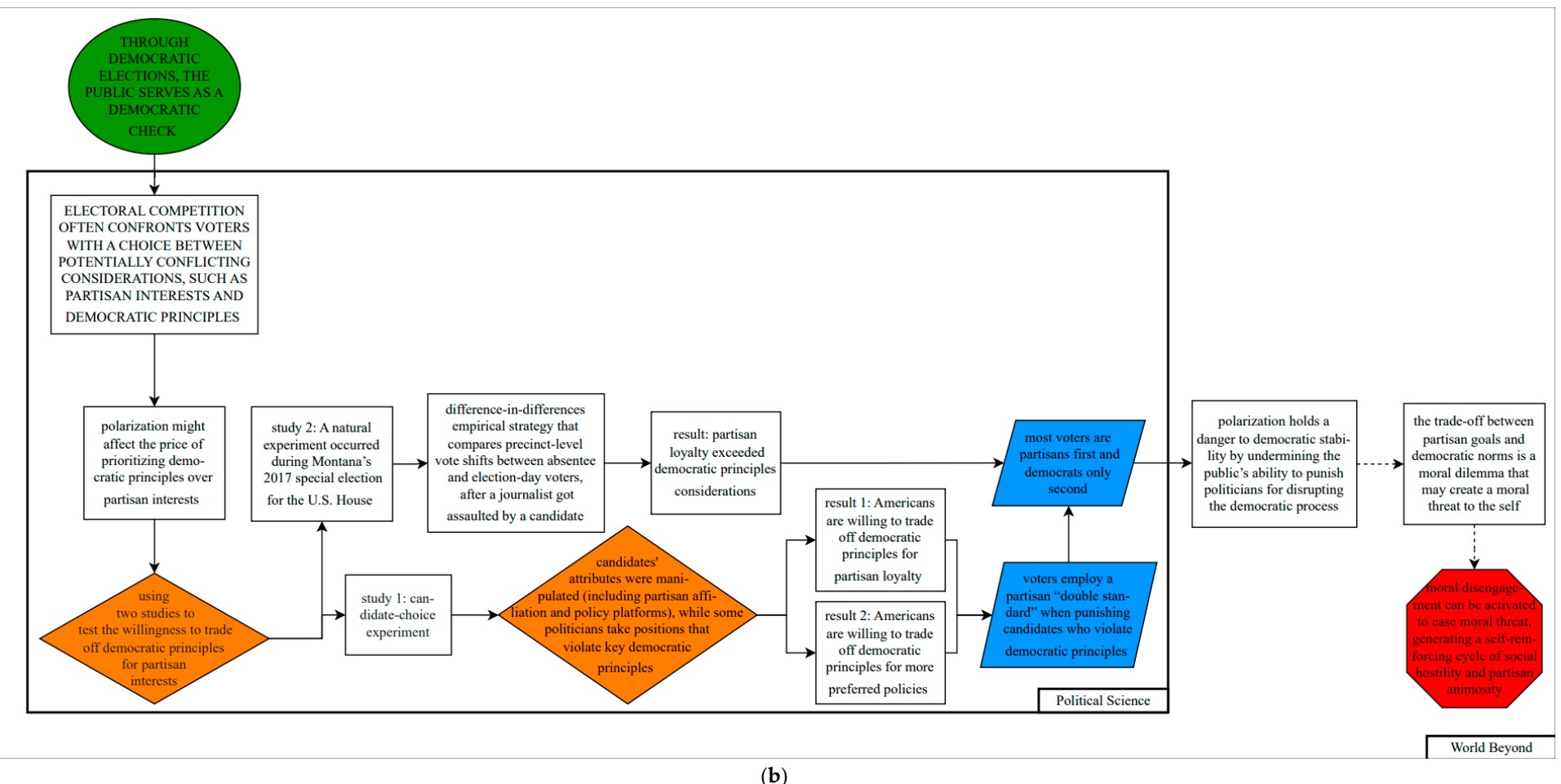

(**b**)

**Figure 3.** (**a**) "Systematic synthesis" diagram of Graham and Svolik's (2020) article. Inspired by Ben-Nun Bloom et al.'s (2020) theoretical conceptualizations of moral dilemmas as moral threats, the dotted line (symbol for a linkage inferred by the reader) leads to a new terminal variable, suggesting that the trade-off between partisan goals and democratic norms is a moral dilemma that may create a moral threat to the self. (**b**) "Systematic synthesis" diagram of Graham and Svolik's (2020) article. Building on the insights from Ben-Nun Bloom et al.'s (2020) findings, a new terminal conclusion is proposed for Graham and Svolik's (2020) work: similar to how moral disengagement and resentment can be activated to alleviate moral threats in armed conflicts, it can also be employed to mitigate moral threats in partisan conflicts, generating a cycle of social hostility and partisan animosity. Diagrammed by: Pazit Ben-Nun Bloom, Ilona Goldner, Sarah Gansen and Patrick James.

This morally compromising decision between the partisan ingroup goals and personal values is similar to the one presented by Ben-Nun Bloom et al. (2020), where investment in the security of one's ingroup involves harmful or injurious conduct toward an outgroup. As the armed character of the conflict emphasizes the role of conflicts generally in heightening the moral threats and dilemmas involved, we suggested that the conclusion should also include domestic conflicts, like political polarization. The diagram in Figure 4a incorporates this inferred conclusion into the original systemist diagram of the article. The dotted line leads to a new terminal variable: "domestic intergroup conflicts may create heightened moral threats due to moral dilemmas". In this way, our systematic synthesis connects the conclusions of the two articles. We further conclude from this the likelihood that, just as moral disengagement can be activated to ease moral threat in an armed conflict, it can also be used to do so in partisan conflict. This conclusion is the terminal variable of our analysis of Graham and Svolik's work, as illustrated in Figure 3b's diagram.

We wish to highlight two critical implications of associating moral disengagement with harsh acts fueled by affective polarization. First, our analysis reveals that morality plays a dual role in partisan animosity, aiding in its creation and reinforcement, thereby facilitating self-reinforcing cycles of hostility. Such a cycle commences with the moralization of the outgroup, which may generate an existential threat in the political domain, leading to morally questionable behaviors like partisan-based discrimination and support for undemocratic practices (as demonstrated above in Graham and Svolik's work). Subsequently, individuals may adopt moral disengagement as a strategy to reduce moral threats and avoid self-condemnation. A resentment strategy towards out-partisans may emerge, which can deepen and reinforce animosity, thus creating an ongoing cycle of escalation. The escalating animosity may trigger additional negative actions against the opposing party, thus initiating a new cycle and propagating the entrenchment of partisan animus.

This leads to our second inferred critical implication, which concerns conflict resolution in situations where such a cycle of hostility and moral disengagement is present. Our analysis suggested that increasing resentment and repeated difficult moral choices can widen the divide between opposing groups, making reconciliation more difficult. This implication relates to a recurring question in the literature on affective polarization: whether negative actions resulting from polarization stem from ingroup love or outgroup hatred (as claimed, respectively, by (Lelkes and Westwood 2017) and (Abramowitz and Webster 2016)). A recent study by Amira et al. (2021), aiming to resolve this debate, found that the answer depends on the presence of moral threat: when faced with a zero-sum choice between helping their ingroup or harming the outgroup, people tend to choose the former. However, when there is a moral threat to partisan identity, this tendency shifts, and outgroup hatred becomes the predominant choice.

When hatred or hostility becomes the primary driver of actions, the willingness to engage in efforts to obtain a solution seems inadequate. This conclusion has implications far beyond the context of partisan conflict, including that of armed conflicts presented in the article of Ben-Nun Bloom et al. (2020). This new conclusion is the terminal variable of our analysis of Ben-Nun Bloom et al.'s work, as illustrated in Figure 4b's diagram. Due to the possibility of a shift in conflict dynamics from ingroup-focused security concerns to outgroup-driven animosity, the success of peace processes may face greater challenges. Such a shift would entail a move away from resolving conflicts primarily motivated by concerns for one's own group towards conflicts stemming from hostility toward the outgroups, thereby making peaceful resolutions more difficult.

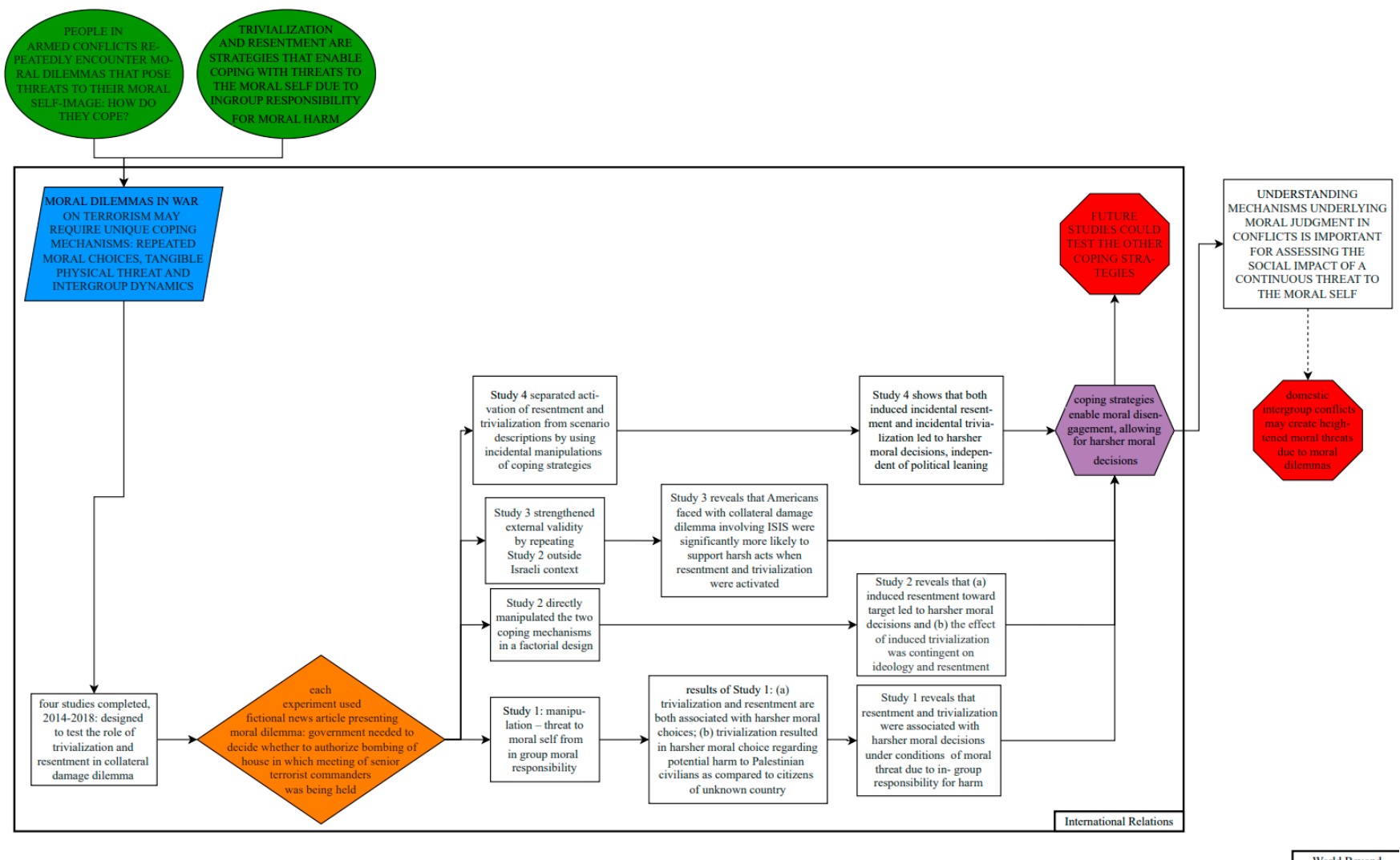

(**a**)

**Figure 4.** *Cont.*

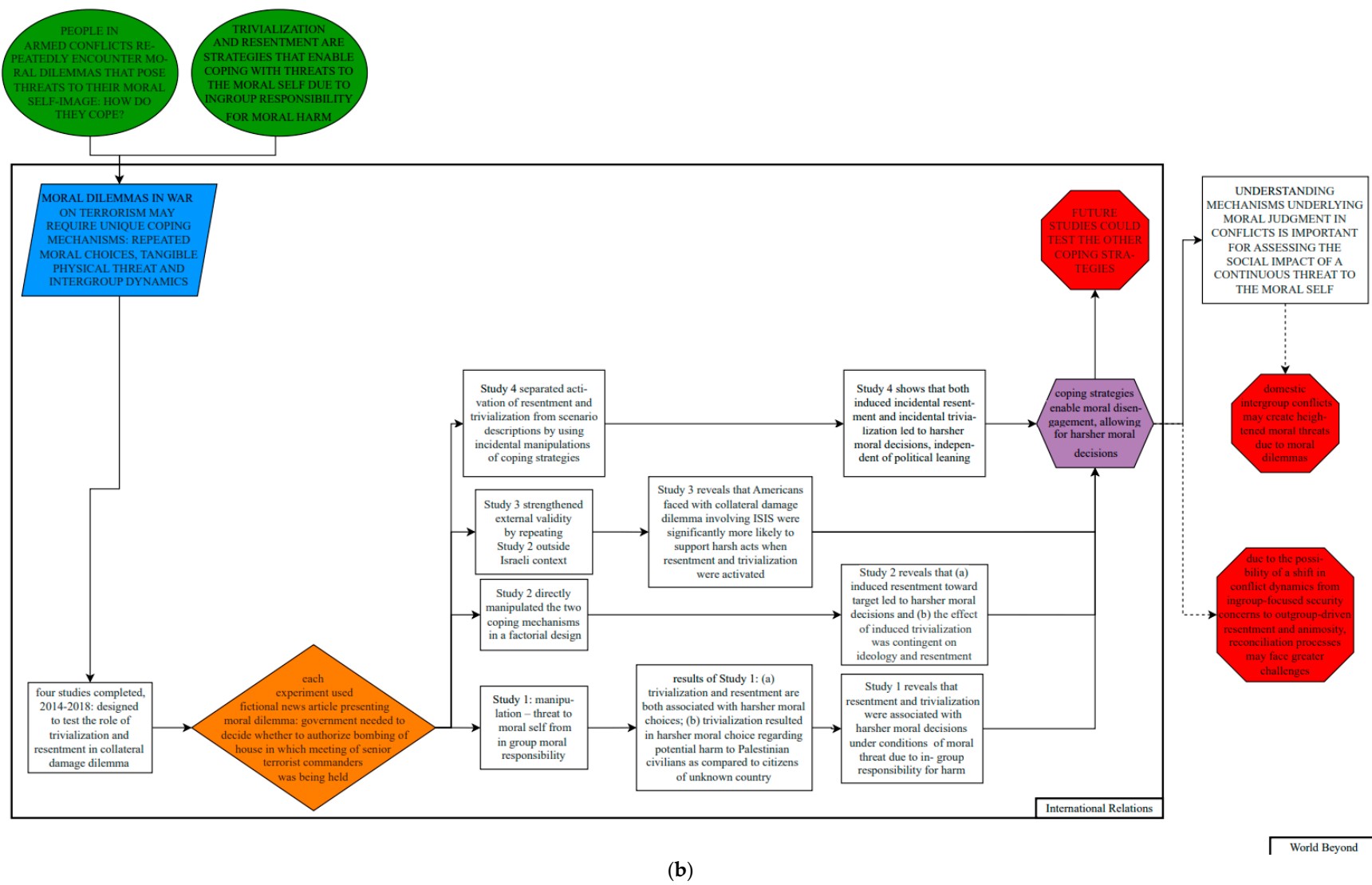

**Figure 4.** (**a**) "Systematic synthesis" diagram of Ben-Nun Bloom et al.'s (2020) article. Based on the dilemma presented in Graham and Svolik's (2020) article, which revolves around the tension between personal democratic norms and ingroup partisan goals, the diagram includes a dotted line (representing a connection inferred

by the reader) that leads to a new terminal variable. This suggests that domestic intergroup conflicts, like international conflicts, might generate heightened moral threats due to moral dilemmas. (**b**) "Systematic synthesis" diagram of Ben-Nun Bloom et al.'s (2020) article. Drawing on the body of literature concerning partisan animosity, particularly the examination of whether negative actions arising from polarization are driven by ingroup love or outgroup hatred, our analysis draws a new conclusion regarding the significance of the coping mechanism of resentment. This conclusion suggests that resentment may alter the conflict dynamic towards outgroup hate, thereby making reconciliation more difficult in both domestic and international conflicts. Diagrammed by: Pazit Ben-Nun Bloom, Ilona Goldner, Sarah Gansen and Patrick James.

## 5. Conclusions

In our analysis, systemist graphics allowed for a deeper understanding of the cause-effect relationships posited or inferred in each article, and highlighted the connections and trends between them. By using this method, our analysis uncovered the shared social phenomenon of moralization in conflicts. This allows us to suggest advances in the fields of both polarization and conflict resolution. Adding to our understanding of the interactions within and between the systems under study, the method provides new insights into how social and political phenomena manifest and interact. In the current article, despite the commonly held belief that morality is associated with the social good and caring for others, our analysis has uncovered the intricate ways in which moral psychology mechanisms can instead contribute to the *escalation* of social animosity, leading to heightened domestic conflicts such as political polarization and international armed conflicts. We encourage future research to extend beyond the empirical validation of our theoretical framework and incorporate the examination of the political context surrounding conflicts, such as the party system and political culture, as well as individual variations. The present article did not address these critical aspects, despite their potential significance to the dissociation from moralized polarization. For example, while most studies on polarization in the United States do not identify partisan asymmetry in negative sentiments toward political opponents (as also evidenced by Graham and Svolik 2020), the moral psychology literature underscores the contextual dimension as a crucial element for comprehending the influence of coping strategies on moral decision-making. This includes exploring how the moderating effect of political leaning varies across different contexts, as demonstrated by the research of Ben-Nun Bloom et al. (2020). Overall, such differences highlight the urgent need for future interdisciplinary studies that consider the moral implications within societies, particularly in relation to the erosion of moral values and the potential repercussions on liberal democracy, including the risk of democratic backsliding.

**Author Contributions:** Conceptualization, P.B.-N.B. and I.G.; validation, I.G. and P.B.-N.B.; formal analysis, I.G. and P.B.-N.B.; investigation, I.G. and P.B.-N.B.; resources, P.B.-N.B.; writing—original draft preparation, I.G.; writing—review and editing, I.G. and P.B.-N.B.; visualization, I.G. and P.B.-N.B.; supervision, P.B.-N.B.; project administration, P.B.-N.B. All authors have read and agreed to the published version of the manuscript.

**Funding:** This research received no external funding.

**Institutional Review Board Statement:** Not applicable.

**Informed Consent Statement:** Not applicable.

**Data Availability Statement:** Not applicable.

**Conflicts of Interest:** The authors declare no conflict of interest.

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
