# Peer review of "Polarization and Moral Threat: Insights from Systemist Analysis"

_socsci, doi:10.3390/socsci12080453_

Round 1

Reviewer 1 Report

The main thesis of this essay (hereafter, Thesis) is a universal one about affective polarization: that within such polarization, morality plays the role of establishing, catalyzing and reinforcing partisan animosity. Thirteen objections against this article’s defense of this thesis run as follows.

Objection 1: The article aims to support Thesis empirically by relying on findings by Graham and Svolik (2020) and Ben-Nun Bloom et al. (2020). Suppose that these findings are not challengeable. Even in that case, they are not sufficient to back up Thesis. That happens because Graham and Svolik (2020) mainly make cases for particular claims about or related to contemporary USA. Besides also doing that, Ben-Nun Bloom et al. (2020)  also serves to back up claims about contemporary Israel, which the present article does not discuss in detail. Indeed, as the author recognizes (lines 77-78), “the literature on polarization is based on American politics”. So, I think the article should limit itself to make claims about or related to the considerably specific context of this politics. In other words, it is not obvious how findings about this considerably particular context would be helpful to understand how affective polarization works in general, say, in other countries like India, Russia, Germany, South Africa etc.

Objection 2: The article assumes that “democracy” and the predicate, “is democratic” (hereafter, <Democratic>) can be applied loosely, that is, without explicitly stating conditions for using them. This move is problematic. After all, there are disputes on what democracy is and how <Democratic> should be applied. The article appears yet to ignore these disputes while making all sorts of “questionable” and extremely vague claims, such as that “undemocratic practices” (line 310) (in a sense not explained in the article) are “morally questionable”. That is: what kinds of undemocratic practices are at stake here? Is it an undemocratic practice to criticize the way the democratic party ultimately chose Hilary Clinton (as opposed to Bernie Sanders) to run against Trump in 2016? What about opposition to the persecution of Assange? Is it an undemocratic practice to argue that the NYT or CNN seem to do propaganda for the Democrat Party? Etc. The article does not give responses to these questions and several others but seems to insinuate that <Democratic> applies to whatever helps to kept things going as they have been going in the USA.

Objection 3: The article also assumes that <Democratic> is applicable to the current political system of the USA. This is a problematic move, especially if one (like the author) does not care to mention a few characteristics of this system, such as: a) There are only two main parties in this system (the Democratic and the Republican); b) These parties are basically ran by multi-millionaires and even billionaires; c) In practice, it is impossible to run for office without being or having support from these likes; d) Lobby is legal in the USA; e) There is a massive incarceration of black people in this country; f) The USA’s military budget is higher than that of any other country; g) The USA has been constantly involved in wars; h) There are bases of the USA throughout the globe; (i) The inequalities between billionaires and the lower classes of the USA are extremely high; etc.

Objection 4: The article assumes that democracy is under threat in the USA without discussing other plausible alternative theses: namely, that there never was a “democracy” in the USA; that instead of opposing democracy, the American people is starting to realize that democracy never existed in the USA; that the press has never been free in the USA (as indicated by Assange’s and Snowden’s struggles); that <Democratic> is not attributable to any American president, to the USA’s elections, the USA’s war-driven attitudes vis-à-vis its opposing states etc.

Objection 5: The article does not contextualize the thesis that the USA’s democracy is under threat in explicitly acknowledging that this thesis only started to be defended after Trump’s election in 2016.

Objection 6: The article does not even mention the distinction between republicans and democrats as if this distinction were not important vis-à-vis an article that tackles American politics.

Objection 7: In following Graham and Svolik (2020), the article does not care to develop other forms of empirical research, e.g., that on whether, to begin with, those who are “willing to trade off democratic principles for either partisan loyalty or preferred politics” (139-140) truly believe that democratic principles (e.g., “the press should be free”) have actually been applied in the USA.

Objection 8: The article assumes that “morality” and the predicate, “is moral” (hereafter, <Moral>) can be applied loosely, that is, without explicitly stating conditions for using them. This is another problematic move. As any introductory class or book on ethics indicates, there has never been agreement on how “morality” and <Moral> should be used. Consider Aristotle, Aquinas, Kant, the utilitarians, Marx, Nietzsche, Rawls etc… These philosophers apply “morality” in all kinds of senses. I could not understand which one (if any) was adopted by the article.

Objection 9: Given objection 8 and the fact that Thesis uses the term, “morality”, Thesis is an extremely vague thesis.

Objection 10: Given objection 8, it is unclear what is meant by the article’s problematization of the view that “moral character promotes the good” (349). After all, the traditional philosophical term, “good”, is another one that the article applies loosely, regardless of the fact that any introductory class or book on ethics indicates that one should not do that.

Objection 11: Indeed, depending on how “morality” is understood, Thesis is trivial.

Objection 12: Granted the article loose use of “morality”, all sorts of theses also follow. For instance, that morality also plays a role in disrupting and diminishing partisan animosity; that morality ultimately plays a role in everything related to human beings etc.

Objection 13: In lines 341-342, the article states that “systemist graphics allowed for a deeper of the cause-effect relationships posited or interred in each article”, that is, Graham and Svolik (2020) and Ben-Nun Bloom et al. (2020). This is not a very convincing claim. These articles resort to a quite straightforward language while explaining their findings in considerably clear fashions. So it is not clear that the graphics help to clarify such findings, especially considering that the “full explanation of the notation of systemist graphics” (lines 33-34) only appears in a forthcoming paper that was not available to me. For instance, I did not get the role of the colors in such graphics as well as the role of its distinct geometrical forms, such as circles and squares. But even if the forthcoming paper were available, it is problematic to argue that these graphics are useful if before reading them, one needs to read a specific notation that may be harder to understand than the upfront language of the articles these graphics are supposed to clarify and summarize.

Reviewer 2 Report

The article deals with a highly relevant topic, especially in the current time. Its based on an intensive study of the state of research on affective polarization and moral thread, elegantly presented, contextualized and interrelated in point two, the authors succeed in leading to their research interest: What role morality plays in polarization. In order to visualize the relationships between the different variables concerning the study, they use systematic visual representation, an innovative way of representing relationships and epistemological process. By linking the content of two, as the authors themselves say, "profoundly dissimilar" articles, the paper can also be seen as an important stimulus to increased interdisciplinary research and linkage. I congratulate the authors on this useful contribution to the research literature and have no substantive criticisms to offer. 

Reviewer 3 Report

The draft is highly theoretical and heavily leaning to abstract statements, with little, if any, empirical evidence to support the narrative. Additionally, and most importantly, the draft was submitted to the Section International Migration, Special Issue The Visual International Relations Project - and It does not seem to fit thematically to either of these.

However, I would like to thank the authors for looking into the intricate ways, in which moral psychology mechanisms may contribute to the escalation of social animosity. I would be curious to read about the application of theoretical findings into concrete context, where the mechanisms under study led to heightened domestic conflicts - or international armed conflicts.

Reviewer 4 Report

This article on ‘Polarization and Moral Threat’ is well written and quite interesting. I had not before connected these literatures and learned a lot. This of course is the core goal of the article using the systemist approach. This is so well done I am not able to garner suggestions.

Round 2

Reviewer 1 Report

Reply to reply 1: The authors seem to run into a tension or even a contradiction; that their article is a (a) “theoretical contribution” and that yet (b) its thesis should “be directly and empirically validated”. This is confusing. The authors do not seem to rely on a reasonable take on how the theorical / empirical distinct is to be drawn.

Reply to reply 2: It does not suffice to add the expression, “liberal”, in front of “democracy” to avoid my previous Objections 2 to 5. I reject the paper’s assumed use of the term, “democracy”, while tending to likewise reject an assumption of the paper: that “democracy” exists in the USA.

Reply to reply 3: The authors say that the discussion of the current political system of the USA is beyond their scope. Ok. It is also beyond this reply’s scope to justify the views: that this article may as well be a symptomatic expression of this system; and that perhaps the democracy score system of Freedom House should not be taken very seriously, given that “democracy” does not seem to be a thing that can be precisely counted.

Reply to reply 4: Regardless of what Graham and Svolik do, the distinction between Democrats and Republicans, as my previous Objection 6 indicates, is relevant for this essay. That is so because its theses are quite similar to what Democrats usually believe and also quite dissimilar from Republicans’ views. (This is not to state that I am sympathetic to the latter).

Reply to reply 5: The fact that a practice is widely shared in a field (e.g., moral psychology) is not sufficient reason for taking it for granted. If the paper aims to remain neutral on distinct notions of “morality” while only considering what particular individuals take morality to be, then, it should be much more careful regarding its use of terms, such as “moral”, “good”, “bad” etc. For instance, the paper’s core thesis that “morality plays a two-fold role in affective polarization” should be rearticulated.

Reply to reply 6: I do not see how the comments the authors make to what they call comment 7 are particularly pertinent vis-à-vis my Objections 7 and 9 to 12.

Reply to reply 7: I still do not see how this paper makes a useful use of systemist graphics.

Author Response

We would like to express our sincere appreciation to the editors and the second-round reviewer for their valuable feedback, which we have addressed in the following letter and through revisions in the manuscript.

In response to the first request of the editors, we have incorporated a definition of democracy in two key sections of our article: at line number 63 for a basic definition and at lines 100 for a comprehensive one.